# Proteasome-Mediated Regulation of GATA2 Expression and Androgen Receptor Transcription in Benign Prostate Epithelial Cells

**DOI:** 10.3390/biomedicines10020473

**Published:** 2022-02-17

**Authors:** Waqas Azeem, Jan Roger Olsen, Margrete Reime Hellem, Yaping Hua, Kristo Marvyin, Xisong Ke, Anne Margrete Øyan, Karl-Henning Kalland

**Affiliations:** 1Department of Clinical Science, University of Bergen, 5021 Bergen, Norway; jan.roger.olsen@sveio.kommune.no (J.R.O.); margrete.reime.hellem@helse-bergen.no (M.R.H.); yaping.hua@uib.no (Y.H.); marvyin@gmail.com (K.M.); xisongke@hotmail.com (X.K.); anne.oyan@uib.no (A.M.Ø.); 2Centre for Cancer Biomarkers, University of Bergen, 5021 Bergen, Norway; 3Department of Microbiology, Haukeland University Hospital, 5021 Bergen, Norway; 4Department of Immunology and Transfusion Medicine, Haukeland University Hospital, 5021 Bergen, Norway

**Keywords:** prostate cell, luminal differentiation, androgen receptor, GATA2, proteasome, MG132

## Abstract

GATA2 has been shown to be an important transcription factor together with androgen receptor (AR) in prostate cancer cells. Less is known about GATA2 in benign prostate epithelial cells. We have investigated if GATA2 exogenous expression in prostate epithelial basal-like cells could induce *AR* transcription or luminal differentiation. Prostate epithelial basal-like (transit amplifying) cells were transduced with lentiviral vector expressing GATA2. Luminal differentiation markers were assessed by RT-qPCR, Western blot and global gene expression microarrays. We utilized our previously established AR and androgen-dependent fluorescence reporter assay to investigate AR activity at the single-cell level. Exogenous GATA2 protein was rapidly and proteasome-dependently degraded. GATA2 protein expression was rescued by the proteasome inhibitor MG132 and partly by mutating the target site of the E3 ligase FBXW7. Moreover, MG132-mediated proteasome inhibition induced *AR* mRNA and additional luminal marker gene transcription in the prostate transit amplifying cells. Different types of intrinsic mechanisms restricted GATA2 expression in the transit amplifying cells. The appearance of *AR* mRNA and additional luminal marker gene expression changes following proteasome inhibition suggests control of essential cofactor(s) of *AR* mRNA expression and luminal differentiation at this proteolytic level.

## 1. Introduction

Most prostate cancers present with a phenotype of predominantly luminal epithelial cell markers as opposed to basal cell markers [1]. In our model of stepwise tumorigenesis of human benign prostate epithelial cells, the tumorigenic cells lack prostate luminal cell differentiation markers [2,3,4,5]. The stepwise tumorigenesis model was derived from EP156T cells which are benign human prostate basal epithelial cells immortalized by hTERT [6]. Stepwise tumorigenesis of EP156T cells was achieved based upon physiological growth selection over extended times [7] and epithelial to mesenchymal transition (EMT) was an early step during selection [3]. We have previously found that the transcription of the androgen receptor (*AR*) mRNA is effectively silenced in EP156T cells, and luminal differentiation markers and AR target genes such as *KLK3, KLK2, FKBP5* and *NKX3-1* are not induced by androgen addition [8]. These AR target genes are, however, readily induced by androgen following exogenous expression of AR, but with endogenous AR remaining silent [8]. In addition to being a master regulator of the luminal cell secretory program, AR appears to be essential in the regulation of basal to luminal cell reprogramming [9,10] with critical aberrant reprogramming in prostate cancer [11]. The molecular mechanisms of *AR* mRNA expression and repression in benign prostate cells remain unknown [12]. Here, we investigated the potential of the pioneering transcription factor GATA2 to induce *AR* mRNA expression and luminal AR target genes in our EP156T cell culture model. In order to substantiate our investigations, we altogether employed four different benign prostate epithelial cell types: RWPE-1 cells [13], hTERT-immortalized EP156T cells [6]; hTERT-immortalized 957E/hTERT cells [14,15] and non-immortalized primary prostate cells (PrEC). All four cell types express KRT5, KRT14, KRT8 and KRT18 [16,17]. We, therefore, consider them transient amplifying (TA) cells with basal cell features and hereafter refer to them collectively as TA cells. Interestingly, different transcription factors exhibited different types of restricted expression in TA cells. GATA2 was highly susceptible to proteasomal degradation in all benign TA cells in contrast to in LNCaP and 22Rv1 prostate cancer cell lines. Proteasome inhibition using MG132 could rescue GATA2 protein, but additionally led to the unexpected and serendipitous discovery that proteasome-regulated co-factors could be essential to induce *AR* mRNA expression and additional luminal marker type gene expressions in TA cells.

## 2. Materials and Methods

### 2.1. Cell Lines and Cell Cultures

Immortalized 957E/hTERT and 957E/hTERT-AR human prostate epithelial cells were generously provided by John T. Isaacs lab (Johns Hopkins, Baltimore, MD, USA) [9,10]. The prostate epithelial cell line RWPE-1 was purchased from the American Type Culture Collection (ATCC, Cat. No. CRL-11609). The cells, 957E/hTERT, 957E/hTERT-AR and RWPE-1, were grown in Keratinocyte-SFM (serum-free medium) supplemented with bovine pituitary extract (BPE) and human recombinant epidermal growth factor (EGF) supplied as a kit (Cat. No. 17005-042, Thermo Fisher Scientific, Waltham, MA, USA), and 1% penicillin-streptomycin. Primary prostate epithelial cells (PrEC) were purchased from ATCC, Manassas, VA, USA (Cat. No. PCS-440-010) and were grown in prostate epithelial cell basal medium (ATCC, PCS-440-030), supplemented with prostate epithelial cell growth kit contents (ATCC, PCS-440-040). The prostate epithelial cell line EP156T was grown in MCDB153 medium (Biological Ind. Ltd., Beit Haemek, Israel), supplemented with 1% fetal calf serum (FCS), growth factors and antibiotics as described [3,5]. Complete growth media were changed every 2–3 days.

### 2.2. Vectors, Lentiviral Particles, Transfection and Transduction

Homo sapiens GATA binding protein 2 (*GATA2*) wild-type (wt), transcript variant 1 (NM_001145661.1) lentiviral expression vector (Cat. No. EX-F0016-Lv205) and *GATA2* T176A mutant lentiviral expression vector (Cat. No. CS-F0016-Lv205-01) were designed and constructed in collaboration with Labomics S.A. (Nivelles, Wallonia, Belgium). Lentiviral particles were produced by the transfection of the expression vector in HEK 293Ta lentiviral packaging cells following the manufacturer’s protocol (Cat. No. HPK-LvTR-20, GeneCopoeia, Rockville, MD, USA). The lentiviral particles from the supernatant were concentrated using Lenti-Pac Lentivirus Concentration Solution (Cat. No. LPR-LCS-01, GeneCopoeia), following the manufacturer’s instructions. Harvested lentivirus was transduced in the presence of polybrene at the final concentration of 6 µg/mL for 24 h. The stably transduced cells were subjected to 6 µg/mL puromycin, unless otherwise stated.

During initial parts of this work, wild-type GATA2 was expressed based upon *GATA2* lentiviral particles (Cat. No. OHS5900-202621424), and together with Precision LentiORF RFP Positive control plasmid (Cat. No. OHS5832), were bought from Dharmacon (Lafayette, CO, USA).

### 2.3. RNA Interference

Short interfering RNAs (siRNAs) for *FBXW7*, *si-FBXW7* (s30665) and control siRNAs were purchased from Ambion Austin, TX, USA (Cat. No. 4392420, Silencer Select Pre-Designed siRNA). The siRNA transfections were performed using a Lipofectamine RNAiMAX transfection reagent (Cat. No. 13778075, Invitrogen, Waltham, MA, USA), according to the manufacturer’s protocol. The nucleotide sequence of *si-FBXW7* was 5′-CGGGUGAAUUUAUUCGAATT-3′.

### 2.4. Western Blotting

Western blots (Wb) were performed as previously described [8]. The primary antibodies used were as follows: Anti-GAPDH (Cat. No. ab181602, Abcam, Cambridge, UK), anti-FBXW7 (Cat. No. ab109617, Abcam, Cambridge, UK), anti-beta-actin (Cat. No. ab8226, Abcam), anti-AR (Cat.no. ab133273) and anti-GATA2 (Cat. No. sc-515178, Santa Cruz Biotechnology, Dallas, TX, USA). The horseradish peroxidase (HRP)-conjugated secondary antibodies used were as follows: Anti-rabbit (Cat. No. NA934, Amersham, Buckinghamshire, UK), anti-mouse (Cat. No. 170-501, Bio-RAD, Hercules, CA, USA) and anti-goat (Cat. No. P0160, DAKO, Glostrup, Denmark).

### 2.5. Real-Time (RT) Reverse Transcription Quantitative PCR (qPCR)

Total RNAs were extracted using the miRNeasy minikit (Cat. No. 217004, Qiagen, Hilden, Germany), following the manufacturer’s protocol. During total RNA purification, on-column DNase digestion was performed using RNase-Free DNase set (Cat. No. 79254, Qiagen). ss-cDNA was synthesized and RT-qPCR was performed as described previously [18], using pre-designed Taqman probes (Thermo Fisher Scientific, Waltham, MA, USA) with the following assay ID numbers: *AR* (Hs00171172_m1), *GATA2* (Hs00231119_m1), *FBXW7* (Hs00217794_m1) and *ACTB* (Hs99999903_m1).

### 2.6. Agilent Microarrays

The Agilent Human Whole-Genome (4x44 k) Oligo Microarray with Sure Print Technology (Agilent Technologies, Palo Alto, CA, USA, Design. No. G4845A) was used to analyze samples in the present study. A minimum of 3 biological replicates were used per condition. Total RNA purification, cDNA labelling, hybridization and normalization of microarrays have been described previously [4,18]. Following Loess normalization, significance analysis of microarray (SAM) of the J-Express program (2012 version) package (http://www.molmine.com) was used for identification of differentially expressed genes [19]. For interpreting gene expression data, Gene set enrichment analysis (GSEA) was performed using J-Express with default settings. The gene set file for gene ontology terms used was: “c5.all.v6.1.symbols.gmt”, downloaded from http://software.broadinstitute.org/gsea/index.jsp (accessed on 15 January 2018). The ArrayExpress accession number is E-MTAB-8487 (https://www.ebi.ac.uk/arrayexpress/experiments/E-MTAB-8487/; accessed on 18 February 2020).

### 2.7. RNA Sequencing (RNA-seq)

The RNA-seq data were generated and analyzed as published previously [8]. RNA-seq data are available at Gene Expression Omnibus (ID: GSE71797).

### 2.8. Statistical Analysis

All data were analyzed using GraphPad Prism 9 (GraphPad software, San Diego, CA, USA) if not stated otherwise. Statistical significance of the difference was calculated using two-way analysis of variance (ANOVA) followed by Sidak’s multiple comparisons test with 95% confidence interval. A value of *p* ≤ 0.05 was considered statistically significant.

Results from the real-time quantitative RT-qPCR were analysed by RQ Manager v1.2 and DataAssist v3.01 software (Applied Biosystems, Waltham, MA, USA). The expressions of the target genes were normalized to *ACTB* (β-actin) to calculate relative quantification (RQ) of the genes of interest. Error bars represent the standard error of the mean expression level (RQ) based on the RQmin/max of 95% confidence level.

## 3. Results

### 3.1. Expression of GATA Factors in Prostate Cell Lines

To exploit our exploratory RNA-seq analysis data, we wanted to investigate GATA transcription factors expressed in different prostate cell lines, including the benign basal-like epithelial EP156T, the mesenchymal malignant EPT3-M1 which is derived from EP156T cells, and the three cancer cell lines LNCaP, 22Rv1 and VCaP in the absence or presence of synthetic androgen R1881 [8].

*GATA2* expression was very low in EP156T cells and EPT3-M1 cells, whereas the most expressed GATA factors in these cells were *GATA3* and *GATA6*, respectively. In the three established cancer cell lines, LNCaP, 22Rv1 and VCaP, *GATA2* was the predominant GATA factor, although the expressions in 22Rv1 of both isoforms and transcript variants were lower than in the two other cell lines. In line with prior evidence pointing to an important role of GATA2 for AR target gene expression, 22Rv1 cells expressed significantly lower levels of *KLK3* than LNCaP and VCaP. However, 22Rv1 cells expressed some *GATA3* in contrast to LNCaP and VCaP cells (Table 1).

### 3.2. Exogenous GATA2 Was Degraded in TA Cells

To investigate the role of GATA2 on AR expression, GATA2 was exogenously expressed using transduction of TA cells [15,20]. *GATA2* mRNA was undetectable in 957E/hTERT (Figure 1A) and weakly expressed in EP156T cells (Appendix A), consistent with the RNA-seq analysis, indicating that its expression was very low or absent in prostate epithelial cells with a basal-like phenotype (TA cells). Interestingly, although *GATA2* mRNA expression in GATA2-transduced 957E/hTERT (Figure 1A) and EP156T cells (Appendix A) as measured by qPCR was higher than in LNCaP cells, GATA2 protein expression was low or absent when repeatedly tested (Figure 1B and Appendix A). No *AR* was detected using RT-qPCR or Western blot in 957E/hTERT-GATA2 cells (Appendix A).

It has previously been shown that GATA2 is rapidly degraded in some cell types [21]. We treated cells with the proteasome inhibitor MG132 to test the possibility that GATA2 was proteasome dependently degraded in TA cells. Treatment of 957E/hTERT-GATA2 cells with MG132 indeed led to strikingly higher GATA2 expression (Figure 1C). In order to evaluate if higher levels of GATA2 were able to induce the expression of *AR* mRNA, we again performed qPCR. Surprisingly, *AR* was now detectable, albeit at a low level. However, *AR* mRNA was detectable at a similar level in MG132-treated non-transduced control cells suggesting that other factors than increased GATA2 protein were responsible for *AR* mRNA appearance (Figure 1D). We further wanted to assess if the degradation of GATA2 also occurred in prostate cancer cells that rely on GATA2 for proper AR transcriptional function [22], or if GATA2 degradation may be a mechanism of protection from AR activity/differentiation in basal-like prostate epithelial cells. Interestingly, neither LNCaP nor 22Rv1 prostate cancer cells showed substantially increased levels of endogenous GATA2 protein after treatment with the proteasome inhibitor MG132 (Figure 1E,F). We next tested whether GATA2 degradation may be a protection mechanism against exogenous GATA2. However, exogenous AR expressed well in transfected HEK 293FT cells, and MG132 treatment did not lead to further substantial increase of exogenous GATA2 accumulation (Figure 1G).

### 3.3. Proteasome-Mediated Degradation of GATA2 in TA Cells

It has previously been reported that GATA2 levels are controlled by proteasome-mediated degradation in different cell types [23]. An E3 ligase, FBXW7, has been identified for ubiquitin-dependent degradation of GATA2. FBXW7 binds to an amino acid sequence motif containing phosphorylated threonine (Thr) and ubiquitinates GATA2. The substitution of GATA2 Thr176 with alanine suppressed GATA2 degradation [24]. We next investigated whether Thr176 to alanine (A) substitution could stabilize GATA2 protein levels in TA cells. For this purpose, *GATA2* T176A mutant was derived from the *GATA2* wt vector and different prostate basal-like TA cells were transduced. Cells were split once after 48 h of transduction and then treated with MG132 for 8 h before being harvested. In Western blots, GATA2 wt and GATA2 T176A proteins showed signs of degradation before being rescued by the treatment of MG132 in 957E/hTERT cells (Figure 2A). Interestingly, *GATA2* wt and *GATA2* T176A mRNAs were significantly higher expressed in MG132-treated cells compared to non-treated cells (Figure 2B). Moreover, although *GATA2* wt mRNA expression was higher than the *GATA2* T176A mRNA following transduction in 957E/hTERT cells, GATA2 wt protein showed more proteasome-mediated degradation.

These results were also tested in the additional available prostate TA cells. In EP156T cells, exogenous GATA2 was degraded, but was rescued by MG132 treatment. Interestingly, GATA2 T176A showed very little or no degradation with or without MG132 treatment (Figure 2C). The mRNA expression of *GATA2* wt and *GATA2* T176A showed a significant increase in expression when treated with MG132 as compared to non-treated cells (Figure 2D).

Similar findings were observed when RWPE-1 and PrEC cells were transduced with *GATA2* wt or *GATA2* T176A. The degradation of GATA2 wt protein was decreased with MG132 treatment, whereas GATA2 T176A protein only showed a very small or no increase after MG132 treatment (Figure 2E,G). Consistent with our previous findings, a significant increase in mRNA expression of *GATA2* wt or *GATA2* T176A was seen (Figure 2F,H).

### 3.4. Proteasome-Mediated Expression of AR mRNA in TA Cells

In order to evaluate if higher levels of *GATA2* T176A with or without MG132 treatment were able to induce the expression of *AR* mRNA, we again performed qPCR. Cells were split once 48 h following transduction with GATA2 wt and GATA2 T176A expression vectors and were treated with MG132 for 8 h before being harvested. Interestingly, *AR* mRNA was detected in 957E/hTERT and EP156T in both non-transduced and transduced cells irrespective of GATA2 wt or GATA2 T176A expression when treated with MG132. Detectable *AR* mRNA was also observed in non-transduced and GATA2 wt transduced RWPE-1 cells, whereas PrEC cells showed a significant increase in *AR* mRNA expression when treated with MG132 in GATA2 wt or GATA2 T176A cells (Figure 3A). The results suggest that GATA2 T176A was not able to induce *AR* mRNA expression; however, MG132-mediated proteasome inhibition rescued *AR* mRNA independent of higher levels of GATA2 T176A in TA cells.

### 3.5. Exogenous GATA2 Induces Luminal Marker Genes in PrEC Cells

Examination of genome-wide microarray gene expression data revealed that MG132 treatment, in addition to *AR* mRNA, induced the substantial increase of endogenous *NKX3-1* mRNA and the substantial decrease of endogenous *TP63* mRNA in immortalized TA cells (Figure 3B). Furthermore, in PrEC cells that were harvested 48 h following *GATA2* wt transduction, we found that several luminal marker genes were significantly induced, and basal markers were reduced when compared to control PrEC wt cells (Figure 4). In addition, gene sets “steroid metabolism and tissue remodeling” and “genes associated with prostate tumorigenesis” were also significantly up-regulated in PrEC-GATA2 wt cells (Figure 4).

### 3.6. Reduction in Endogenous FBXW7 Stabilized Exogenous GATA2 Protein

Next, we investigated whether the knockdown of *FBXW7* could stabilize GATA2 protein in TA cells. For this purpose, we used several siRNAs for *FBXW7* and tested for optimal concentration and duration of transfection (Appendix A). In knockdown experiments using different siRNAs for *FBXW7*, the selected *si-FBXW7* (*si-FBXW7-2*) showed more reduction of *FBXW7* in 957E/hTERT cells transduced with GATA2 wt or GATA2 T176A (Appendix A). In order to evaluate whether the depletion of FBXW7 stabilizes *GATA2* mRNA expression, we performed *GATA2* qPCR. As shown in Appendix A, *GATA2* mRNA expression significantly increased with the depletion of FBXW7. In addition to *GATA2* mRNA expression, GATA2 protein was also slightly detected when FBXW7 was depleted using *si-FBXW7-2* in 957E/hTERT cells transduced with GATA2 wt or GATA2 T176A (Appendix A). The results suggest that endogenous *FBXW7* reduction contributed to the stabilization of GATA2 protein in TA cells even though the FBXW7 protein level remained little changed.

## 4. Discussion

AR plays a key role in luminal differentiation and in the maintenance of the luminal secretory program. The mechanisms that silence AR expression before it is switched on as a master regulator of luminal differentiation remain poorly understood but include networking with a set of additional transcription factors [1,9,10,11,12]. Recent work has identified a core set of around 80 genes that are co-regulated by retinoic acid (RA) during early stem cell differentiation to committed basal cells [27,28]. Many of the same genes appear to be co-regulated by androgen-bound AR during the differentiation of basal cells to luminal cells. AR itself is not included in this gene set, and the molecular mechanisms that initiate *AR* mRNA transcription remain a critical and unresolved issue and with possible important implications for the understanding of AR in prostate cancer and prostate CSCs [12]. We have recently published that when AR is exogenously expressed in TA cells, AR target luminal marker genes are readily available for androgen-dependent expression, including a large proportion of the 80 gene set referred to above. However, endogenous AR remained silenced [8]. Here, we tested the performance of the transcription factor GATA2 in the context of benign transit amplifying prostate epithelial cells with basal cell features (TA cells). GATA2, together with FOXA1, are considered pioneer transcription factors with the ability to open up chromatin to induce lineage specific transcription programs. Both transcription factors play important roles in AR-directed transcription in prostate cancer [22,29,30] but have been less investigated in benign prostate cells. In the present study, we found that exogenous expression of GATA2 protein was strongly selected against in TA cells. The restriction of GATA2 expression was predominantly at the level of proteasome-mediated degradation that could be inhibited by the proteasome inhibitor MG132. The same phenomenon was found in additional benign TA prostate cells, i.e., PrEC, EP156T and RWPE-1 cells, but not in LNCaP or VCaP prostate cancer cell lines, in which GATA2 protein levels did not increase substantially due to MG132 treatment. Exogenous GATA2 protein appeared stable in transfected human embryonal kidney HEK 293FT cells. It is possible that GATA2 is selected against in benign prostate cells because it has the potential to induce aberrant luminal differentiation. This was supported by the observation that GATA2-transduced PrEC cells examined shortly following transduction indeed showed gene expression changes associated with luminal differentiation.

It has been published that GATA factors contain a short protein motif dubbed “degron” with a central threonine (Thr176 in GATA2), and this motif can be targeted for ubiquitination followed by proteasome-dependent degradation. FBXW7 has been identified as an E3 ligase that mediates GATA2 ubiquitination [24]. In order to examine if this was the mechanism of GATA2 restriction in prostate TA cells, we designed a new *GATA2* lentiviral vector and modified GATA2-wt with a point mutation to generate *GATA2*-T176A. GATA2-T176A appeared to be more stable than *GATA2*-wt protein in all TA cells tested, i.e., PrEC, 957E/hTERT, EP156T and RWPE-1 cells, and rescue of GATA2-wt by MG132 was more prominent than rescue of GATA2-T176A by MG132. While GATA2 T176A was more stable in PrEC, EP156T and RWPE-1 cells, 957E/hTERT showed signs of degradation suggesting that factors other than FBXW7 may also be responsible for GATA2 degradation.

The results suggest that FBXW7 may function as one of the E3 ligases for the degradation of GATA2. Next, siRNA-mediated knockdown of FBXW7 was attempted with different siRNA sequences in different combinations. Significant knockdown of *FBXW7* mRNA was verified by qPCR. Interestingly, the knockdown of *FBXW7* led to an increase of *GATA2* mRNA, and a slight increase of GATA2 protein in both GATA2-wt and GATA2-T176A cells was observed. These findings show the complexity of factors involved in destabilizing GATA2 at different levels ranging from transcriptional to post-translational levels. Restriction is predominantly due to proteasome-mediated degradation and involves ubiquitination of the target sequence with a central Thr176. FBXW7 E3 ligase seems to contribute to the degradation. The evident, but limited effect of FBXW7 knockdown could be due to insufficient knockdown. In addition, given that more than 800 different E3 ligases exist in mammalian cells, some redundant and overlapping functions can be expected. It is also possible that the T176A mutation does not entirely compromise the degron motif or that additional proteolytic mechanisms contribute. Nevertheless, it can be concluded that GATA2 has restricted expression in benign prostate cells due to proteolytic degradation of GATA2-wt in contrast to the prostate cancer cell lines LNCaP and 22Rv1, and also in contrast to benign human kidney embryonal cells.

One serendipitous discovery in this work was that MG132 treatment induced detectable *AR* mRNA in PrEC, 957E/hTERT and EP156T cells in contrast to in untreated control cells. *AR* was induced to the same level in GATA2 transduced cells and mock transduced cells and non-transduced cells, clearly identifying MG132 treatment as the causative event. The MG132-mediated rescue of AR protein expression in prostate basal cells has been reported previously [26] and mechanisms of AR protein stability have been reviewed in [12]. Moreover, one of the most frequently mutated E3 ubiquitin ligases in prostate cancer, SPOP, reportedly promotes ubiquitination-mediated proteasomal degradation of AR [25]. We are not aware, however, that it has been noted previously that *AR* mRNA and a luminal marker mRNA profile emerge following proteasome inhibition of TA cells. This observation suggests the possibility that one or more co-factors regulated by proteasome-mediated activity restrict *AR* mRNA expression in prostate TA cells and could play a central role in the control of luminal differentiation. It should be noted that exogenous *GATA2* mRNA as well as endogenous *NKX3-1* mRNA was increased in contrast to reduced endogenous *TP63* mRNA in MG132-treated cells. Follow-up work is warranted in order to identify the critical factors involved.

Further attempts to identify critical factors that regulate mRNA transcription and thereby luminal differentiation in TA cell cultures was encouraged by MG132-induced *AR* and *NKX3-1* mRNA expression. This suggested that other explanations than selected mutations or epigenetic repression could be operating in our cell models. Encouragement was offered by the recent publication that exogenous expression of NKX3-1 in RWPE-1 cells was sufficient to induce abundant AR protein expression along with luminal AR target genes and reduced TP63, all of which are hallmarks of luminal differentiation [31]. Unfortunately, we were unable, even with two different NKX3-1 expressor vectors, and with RWPE-1 cells obtained directly from the American Type Culture Collection, to reproduce these events. This was substantiated at the single-cell level in our RWPE-1-NKX3-1 cells by their inability to activate the 241B-ARE-mCherry reporter in the presence of androgen as examined with fluorescence microscopy.

In contrast to exogenously transduced *GATA2*, the *NKX3-1* vector directed both robust mRNA and NKX3-1 protein expressions. This testified to the specific proteasome-mediated restriction of GATA2 expression and ruled out compromised NKX3-1 expression as the explanation of its inability to induce either *AR* mRNA transcription or luminal differentiation in TA cells. Parallel experiments with FOXA1 to be published separately showed that this transcription factor is expressed in TA cells immediately following transduction, but FOXA1-expressing cells were selected against during ensuing passaging, showing one more type of restriction of a transcription factor of importance in prostate homeostasis and cancer progression.

In summary, we have shown that the transcription factor GATA2 exhibits different types of restricted expression in TA cell types. Further investigation of the regulatory mechanisms found in the present work has the potential to improve understanding of physiological prostate homeostasis and aberrant regulation in carcinogenesis, in particular regarding proteasome-mediated regulation of key prostate transcription factors.

## 5. Conclusions

Rapid proteasome-mediated degradation of exogenous GATA2 was found in different benign prostate TA cell types. We found that threonine 176 of GATA2 was important for proteasome-mediated degradation. Although exogenous GATA2 could be rescued by the proteasome-inhibitor MG132, it was difficult to investigate further the physiological role played by GATA2 in luminal cell differentiation. It can be speculated that rapid GATA2 degradation in benign prostate epithelial cells, in contrast to in prostate cancer cells, could be selected as a safeguard against aberrant differentiation of TA cells.

The expression of *AR* mRNA and proteins appears tightly regulated in benign prostate transit amplifying and basal epithelial cells (TA cells). The molecular mechanisms of repressed AR expression remain elusive. The present work has discovered the possibility that proteasome-mediated activity may regulate essential co-factors needed for *AR* mRNA and luminal marker mRNA transcription in prostate TA cells.

## Figures and Tables

**Figure 1 biomedicines-10-00473-f001:**
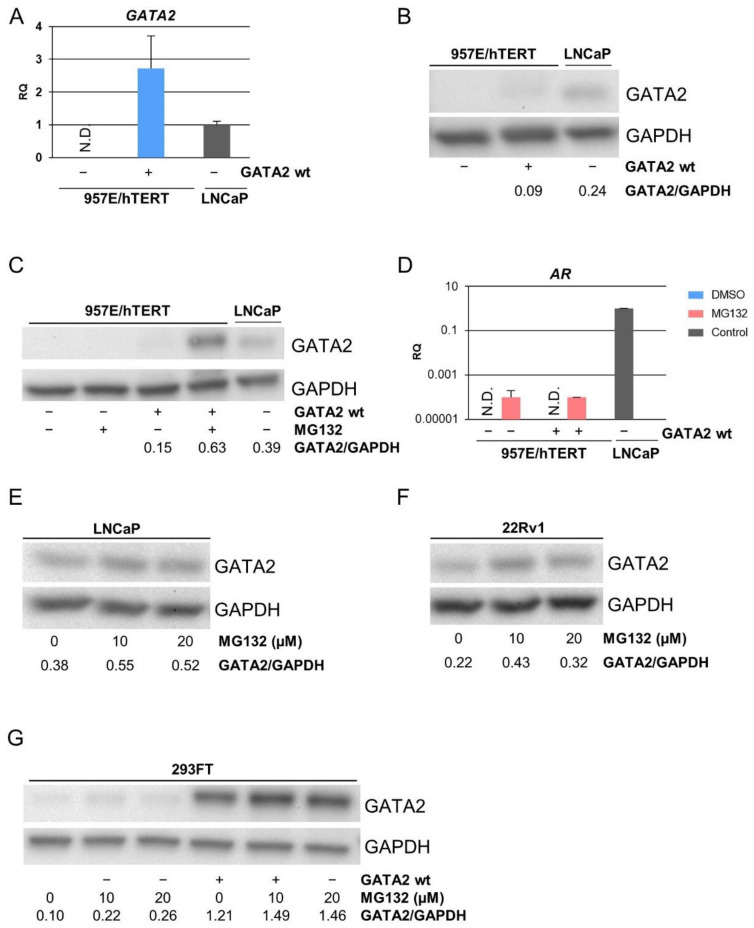
Exogenous expression of GATA2 in basal-like TA cells and GATA2 degradation. (**A**) RT-qPCR and (**B**) Western blot of exogenous GATA2 in 957E/hTERT cells and endogenous GATA2 in LNCaP cells. (**C**) Cells stably expressing GATA2 and control cells treated with DMSO or 10 µM MG132 for 8 h. (**D**) RT-qPCR of 957E/hTERT-GATA2 and control cells treated with 20 µM MG132 for 8 h. (**E**) Western blot of LNCaP and (**F**) 22Rv1 cells treated with indicated concentrations of MG132 for 8 h. (**G**) Western blot of 293FT cells stably expressing exogenous GATA2 or control, and treated with indicated concentrations of MG132 for 8 h. Error bars display 95% confidence intervals. N.D. = not detected.

**Figure 2 biomedicines-10-00473-f002:**
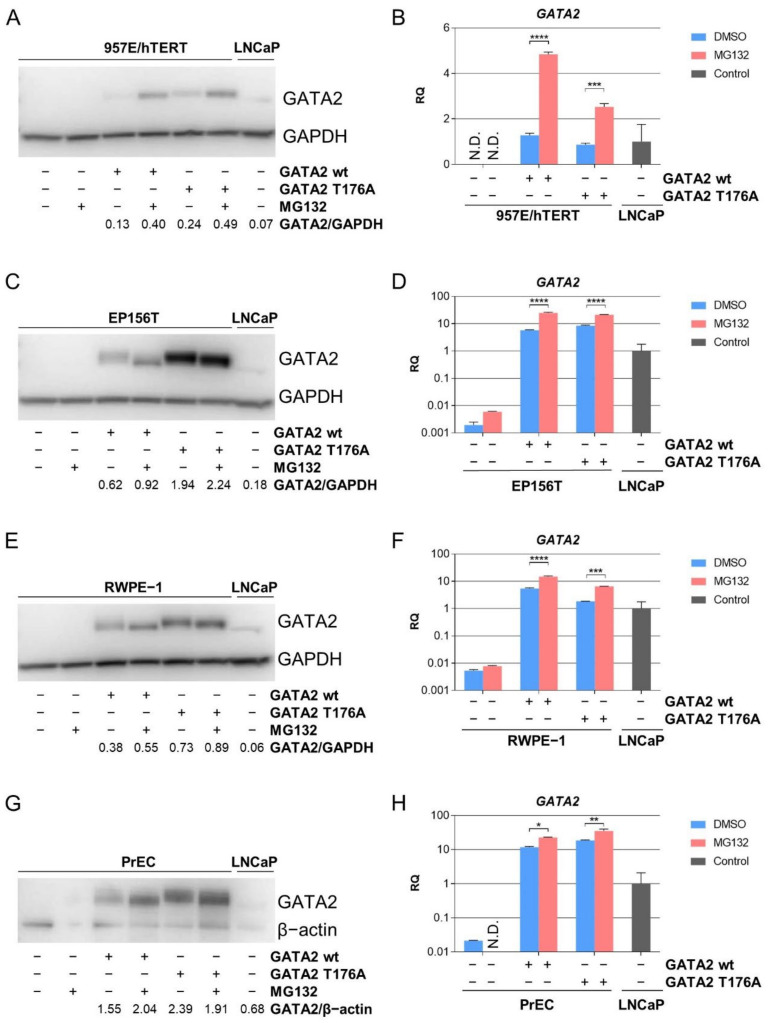
GATA2 proteasome-mediated restriction is present in different TA cells. The basal-like TA cells were split once 48 h post-transduction and treated with 10 µM MG132 for 8 h before being harvested. (**A**) Western blot and (**B**) qPCR of 957E/hTERT cells, (**C**) Western blot and (**D**) qPCR of EP156T cells, (**E**) Western blot and (**F**) qPCR of RWPE-1 cells, and (**G**) Western blot and (**H**) qPCR of PrEC cells. LNCaP cells were used as positive control without MG132 treatment. Error bars represent standard error of the mean expression level (RQ) based on the RQmin/max of 95% confidence level. N.D. = not detected. * *p* ≤ 0.05, ** *p* ≤ 0.01, *** *p* ≤ 0.001, **** *p* ≤ 0.0001 by using one-way ANOVA followed by Sidak’s multiple comparisons test with 95% confidence interval.

**Figure 3 biomedicines-10-00473-f003:**
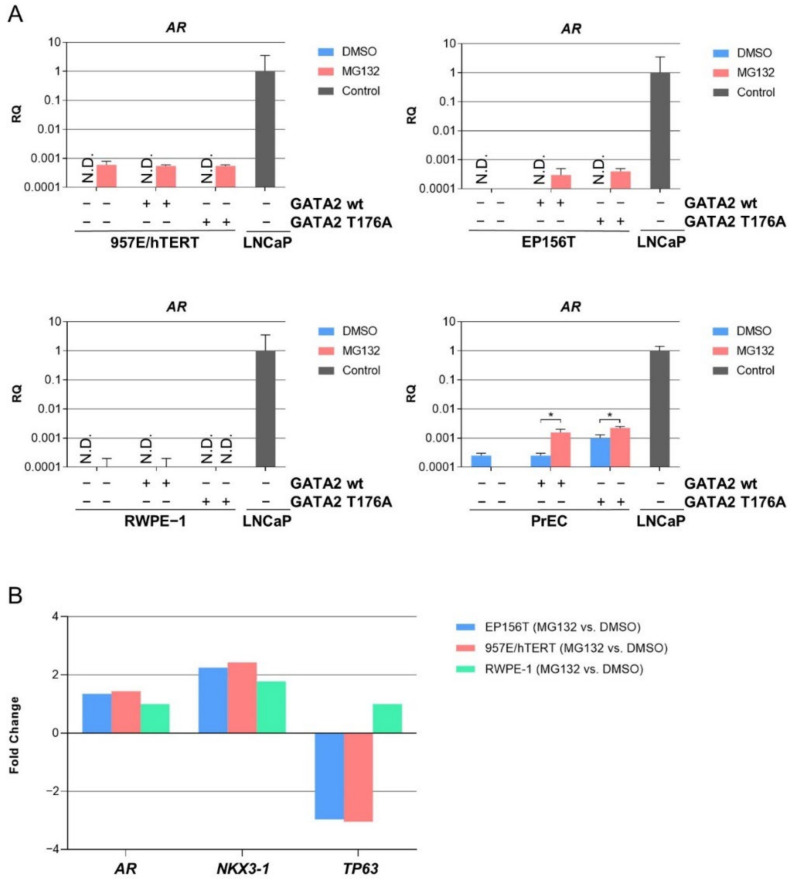
*AR* mRNA proteasome-mediated expression is present in different TA cells. (**A**). The basal-like TA cells were split once 48 h post-transduction and treated with 10 µM MG132 for 8 h before being harvested. qPCR of 957E/hTERT cells, EP156T cells, RWPE-1 and PrEC cells. LNCaP cells were used as positive control without MG132 treatment. Error bars represent standard error of the mean expression level (RQ) based on the RQmin/max of 95% confidence level. N.D. = not detected. * *p* ≤ 0.05 by using one-way ANOVA followed by Sidak’s multiple comparisons test with 95% confidence interval. (**B**) EP156T, 957E/hTERT and RWPE-1 cells were treated with 10 µM of MG132 for 6 h. The corresponding control cells were treated with DMSO. Total RNA was subjected to whole-genome microarray analyses. *AR* was significantly induced only in 957E/hTERT MG132-treated cells (FDR = 0.12) with FDR = 17 in MG132-treated EP156T and RWPE-1 cells. Significant expressions showed FDR = 0.0–0.81 for *NKX3-1* and FDR = 0.0–0.23 for *TP63* in EP156T and 957E/hTERT cells treated with MG312. TP63 was not significantly decreased in RWPE-1 MG132-treated cells (FDR = 25).

**Figure 4 biomedicines-10-00473-f004:**
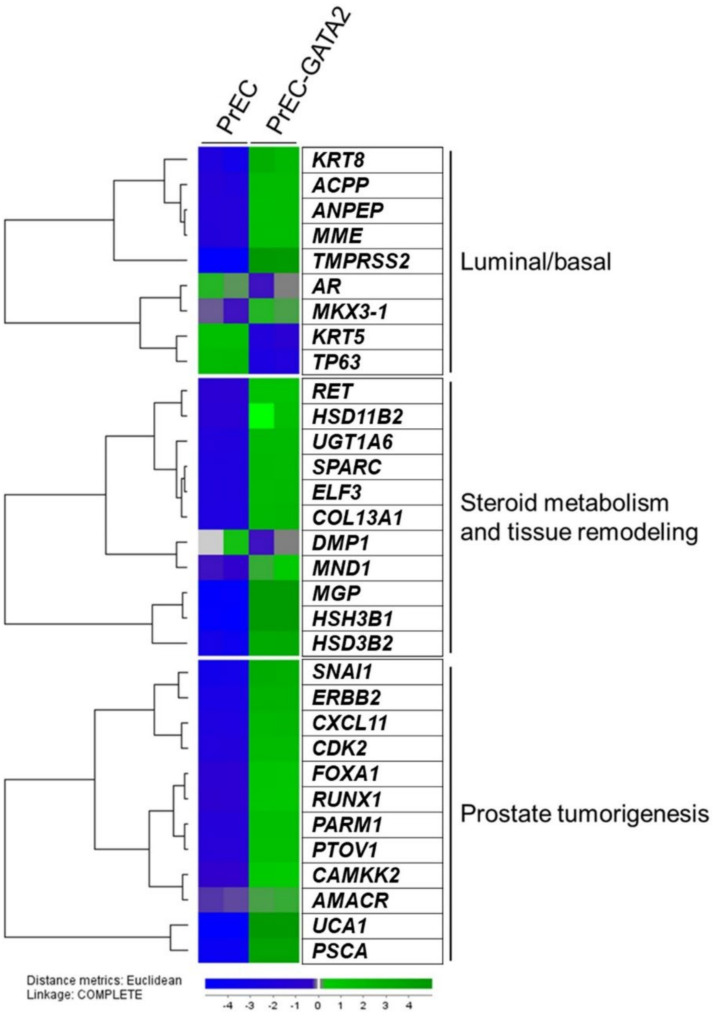
Agilent whole-genome microarray gene expression data. Heat map of selected genes expressed by PrEC transduced with GATA-2 wt. Genes were selected based upon GSEA and c5.all.v6.1.symbols.gmt (Broad Institute). Some presented gene sets like “androgen metabolism” and “tissue remodeling” had few genes in the leading edge. We also used support literature of PubMed and the following references [25,26].

**Table 1 biomedicines-10-00473-t001:** RNA-seq expression of selected genes in prostate benign and cancer cell lines.

	EP156T		EPT3-M1		LNCaP		22Rv1		VCaP	
	Control	R1881 (10 nM)	Control	R1881 (10 nM)	Control	R1881 (10 nM)	Control	R1881 (1 nM)	Control	R1881 (1 nM)
*GATA1*	0	0	0	0	0.1	0	0	0	0	0
*GATA2* isoform 1, transcript variant 1	0.0	0.2	0.4	0.5	6.4	3.6	0.9	0.2	4.5	11.7
*GATA2* isoform 1, transcript variant 2	0.2	0.0	0.1	0.0	14.5	3.6	2.5	1.6	20.2	10.0
*GATA2* isoform 2	0.0	0.0	0	0.0	2.1	0.3	0.7	0.1	3.9	5.5
*GATA3*, transcript variant 1	2.6	1.3	0.3	0.3	0.0	0.0	1.9	0.2	0.0	0.0
*GATA3*, transcript variant 2	1	1	0	0	0	0	0.3	0.5	0	0
*GATA4*	0.0	0.0	0	0.0	0.0	0.0	0.0	0.0	0.0	0.0
*GATA5*	0.0	0.0	0	0.0	0.0	0.0	0.0	0.0	0.2	0.1
*GATA6*	1.4	1.1	2.7	2.9	0.6	0.6	0.8	1.2	1.4	2.1
*KLK3*	0	0	0	0	35	799	4	9	5	52
*NKX3-1*	0	0	4	4	40	139	34	40	171	319

EP156T, EPT3-M1 and LNCaP cells were treated with 10 nM R1881 for 48 h and 22Rv1 and VCaP cells with 1 nM R1881 for 24 h. Values are in fragments per kilobase of exon per million reads mapped (fpkm) and approximated to one decimal point.

## Data Availability

The RNA-seq data presented in this study is openly available at Gene Expression Omnibus (www.ncbi.nlm.nih.gov/geo/query/acc.cgi?acc=GSE71797) under accession number GSE71797 accessed on 9 February 2017. The Agilent whole genome microarray gene expression data presented in this study is openly available in the ArrayExpress database at EMBL-EBI (https://www.ebi.ac.uk/arrayexpress/experiments/E-MTAB-8487/) under accession number E-MTAB-8487 accessed on 18 February 2020.

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
