# Peer review of "Proteasome-Mediated Regulation of GATA2 Expression and Androgen Receptor Transcription in Benign Prostate Epithelial Cells"

_biomedicines, 2022, doi:10.3390/biomedicines10020473_

Round 1

Reviewer 1 Report

All issues have been solved. 

Author Response

We are thankful that we have been able to comply satisfactorily with the reviewer's requests and remain thankful for contributions to improve and clarify this manuscript. Thank you!

Reviewer 2 Report

The authors have modified the manuscript according to some requests. However, some points still remain questionable.

The reviewer agrees upon the fact that there cannot be reasonable doubt about the strong difference between the prostate benign cells and the prostate cancer cells when it comes to GATA2 instability and MG132 rescue. This may be even further emphasized as it is a key finding of the study.

Regarding the novelty of the results shown in Fig 1D, the authors should extend this original finding to the other TA cell lines that were used and analyze AR expression in these models following MG132 treatment in the presence or absence of GATA2 expression. This would generalize the finding to other cell models.

I must say that I do not subscribe to the explanation stating that the statistical differences in mRNA levels between the GATA2 controls (DMSO) and the proteasome inhibitor MG132-treatment that are shown in the graphs of Fig. 2 are highly relevant. It is indeed true for the 957E/hTERT cells but it is much less convincing for the other cell models (especially when mRNA levels are concerned).

Fig. 1G: as stated before the actin loading is far from nice and particularly unequal in terms of loading, which is why quantitation is important in that case. The reviewer thanks the authors for explaining that over-exposition should be avoided but was surprised by the statement that “the authors do not think that quantification is scientifically very useful”.

I have to say that the results of figure 4 are still not conclusive despite the explanations that were provided by the authors. Even if we only consider the effect of siRNA on FBXW7 mRNA levels, this effect is minimal (at most 2.5-fold based on panel A). The same order of magnitude is observed for GATA2 changes which is also not very convincing.

If the authors want to demonstrate the role of FBXW7, they should manage to use more convincing models of FBXW7 repression (Crispr?)

Fig S3 actually shows an increase in FBXW7 as it is labeled in the legend of panel B. (I assume that they used si #2)

What about downregulation of FBXW7 in other TA cell models?

Author Response

Reviewer 2:

The authors have modified the manuscript according to some requests. However, some points still remain questionable.

The reviewer agrees upon the fact that there cannot be reasonable doubt about the strong difference between the prostate benign cells and the prostate cancer cells when it comes to GATA2 instability and MG132 rescue. This may be even further emphasized as it is a key finding of the study.

Regarding the novelty of the results shown in Fig 1D, the authors should extend this original finding to the other TA cell lines that were used and analyze AR expression in these models following MG132 treatment in the presence or absence of GATA2 expression. This would generalize the finding to other cell models.

Author response: We are thankful for these comments and agree that the detection of AR mRNA transcription for the first time in prostate epithelial basal-like transit amplifying cells and perhaps the most important discovery is that MG132-mediated proteasome inhibition rescued AR mRNA independently of GATA2 expression. We have tried to emphasize this observation. This finding has been further extended in other TA cell lines including 957E/hTERT, RWPE-1, EP156T and PrEC cells in Fig 3A where the cells were transfected with GATA2wt or mutant GATA2 T176A and AR mRNA expression was rescued by MG132 independent of GATA2 expression.

Reviewer 2: I must say that I do not subscribe to the explanation stating that the statistical differences in mRNA levels between the GATA2 controls (DMSO) and the proteasome inhibitor MG132-treatment that are shown in the graphs of Fig. 2 are highly relevant. It is indeed true for the 957E/hTERT cells but it is much less convincing for the other cell models (especially when mRNA levels are concerned).

Author response: We do agree that this result is most pronounced for the 957E/hTERT cells. The trend is, however, the same for the additional 3 benign prostate epithelial basal cell types (TA cells). The results were independently reproduced several times (and independently by 2 of the authors of the paper), and for that reason came out as significant according to the statistical analyses. Please also note that the scales on the Y-axes do not emphasize the differences which were around 2-fold or more for all the 4 cell types.

Reviewer 2: Fig. 1G: as stated before the actin loading is far from nice and particularly unequal in terms of loading, which is why quantitation is important in that case. The reviewer thanks the authors for explaining that over-exposition should be avoided but was surprised by the statement that “the authors do not think that quantification is scientifically very useful”.

Author response: We think that the reviewer asks for Fig. 2G in this comment. We acknowledge that the loading controls are unequal and relatively weak, particularly for the LNCaP control cells in this figure. We consequently have complied with the reviewer’s suggested and quantified all the western blots including Fig. 2G in the updated manuscript. The information obtained from the ratios of each lane supports the conclusions drawn. We are thankful for the help to improve the presentation.

Reviewer 2: I have to say that the results of figure 4 are still not conclusive despite the explanations that were provided by the authors. Even if we only consider the effect of siRNA on FBXW7 mRNA levels, this effect is minimal (at most 2.5-fold based on panel A). The same order of magnitude is observed for GATA2 changes which is also not very convincing.

If the authors want to demonstrate the role of FBXW7, they should manage to use more convincing models of FBXW7 repression (Crispr?)

Author response: We thank the reviewer for this admittedly appropriate comment and agree with the reviewer that there may be a room to improve on the knockdown of FBXW7 that can also show a reduction of FBXW7 expression at the protein level. We were not very pleased ourselves regarding the FBXW7 knockdown attempts, but results were consistently reproducible so that additional repeats were futile. Crispr technology is a very good suggestion. As this work has developed, the FBXW7 results have become less central for the main message of this manuscript, and as discussed the experience is that it may be different to obtain very clear results in this model, may be due to redundant back-up mechanisms. For that reason, we have revised this part by moving the Fig. 4A and 4B to Supplementary Fig. 2 in the updated manuscript, thus showing the whole section of FBXW7 knockdown as supplementary information and not as our main results. The microarray gene expression heatmap, previously shown as Supplementary figure 2, is now the new Fig. 4, emphasizing the effect of GATA2 on the induction of luminal marker genes in TA cells.  We are thankful to the reviewer for helping us to make the manuscript clearer and better organized regarding these results.

Reviewer 2: Fig S3 actually shows an increase in FBXW7 as it is labeled in the legend of panel B. (I assume that they used si #2)

Author response: We thank the reviewer for the comment and acknowledge that siFBXW7-1 and siFBXW7-3 didn’t affect FBXW7 mRNA level. Therefore, the Fig. only contained siFBXW7-2 (now called siFBXW7) that decreased FBXW7 mRNA level and significantly increased GATA2 mRNA level.

Reviewer 2: What about downregulation of FBXW7 in other TA cell models?

Author response: We thank the reviewer for this admittedly appropriate comment. siRNAs-mediated knock-down of FBXW7 was also attempted in RWPE-1 cells and found similar effects at the protein level. Hence, 957E/hTERT cell line was used to represent these not so clear, but consistent results in the Supplementary section.

Reviewer 3 Report

In the current study, authors are trying to demonstrate proteasome-mediated GATA2 degradation in an exogenously overexpression system. It is not a good model to use exogenous expression system to study proteasome degradation. Please add AR as a positive control in all the MG132 treatment western blot experiment to show the MG132 treatment efficacy. And, clearly there is not enough evidence showing regulation of GATA2 on AR expression and activity. 

  1. Fig 1. what is the rationale of studying the degradation of exogenous GATA2? Fig 1C, showed exogenous GATA2 protein degradation was E3 ligase dependent, and Fig 1E&F showed the endogenous GATA2 protein level was not increased after treating MG132. This cannot rule out the possibility of due to the exogenous overexpressed GATA2 protein is degraded by E3 ligase and the endogenous GATA2 protein is degraded differently. Fig 1D, increased AR mRNA level in 957E/hTERT cells are very low, and it is not altered by overexpression of GATA2 and treatment of MG132. This is very irrelevant to the hypothesis, please explain the rationale. Fig 1G, it is very hard to argue that it is a protective mechanism to exogenously overexpression of GATA2 which is not increased by MG132 treatment in 293 cells, in vitro. What is the rationale of proposing such mechanism of GATA2?
  2. Fig 2, it is not a good model to study E3 ligase activity in an exogenously overexpression system. And, the MG132 effect on GATA2 wt is not consistent between Fig 2A, and C, E, G, and there is no effect of MG132 on GATA2 T176A in Fig 2A, and C, E, G.
  3. Fig 3, the AR mRNA expression in Fig 3A is too low to have scientific significance.
  4. Fig 4A. the FBXW7 silencing effect should be assayed by Western blot. And, the silencing effect is low in GATA2 wt and T176A overexpressed cells, that could related to the different expression of GATA2 mRNA expression in Fig 4B.

Author Response

Reviewer 3:

In the current study, authors are trying to demonstrate proteasome-mediated GATA2 degradation in an exogenously overexpression system. It is not a good model to use exogenous expression system to study proteasome degradation. Please add AR as a positive control in all the MG132 treatment western blot experiment to show the MG132 treatment efficacy. And, clearly there is not enough evidence showing regulation of GATA2 on AR expression and activity.

Author response: We do not completely agree to this understanding of our work. The initial design was to test if the pioneering transcription factor GATA2, known to interact in the AR transcriptional regulation of prostate cancer, could have the potential to overcome the very tightly regulated repression of AR transcription in benign prostate basal type epithelial cells. We then found that exogenously GATA2 was strongly unstable in all tested of these cell types (TA cells) in contrast to in tested prostate cancer cells and also in contrast to benign non-prostate/ AR-independent cells (HEK293 cells). We only thereafter found that the instability of GATA2 was strongly proteasome-dependent and could be counter-acted by proteasome inhibiton. As a consequence of this experimental development we found that AR transcription was initiated, but was not dependent upon GATA2 expression, but instead was dependent upon proteasome inhibition, with or without GATA2. We consequently have drawn the important conclusion that proteasome-regulated co-factors seem to be crucial in AR transcriptional regulation at the differentiation steps between prostate basal and luminal cells. It should be taken into account that this represents important new insight into an unsolved and very important mechanism of physiological prostate cell differentiation that has been addressed by many research group for many years.

One part of this work was to show molecular details of the GATA2 regulation and we think that we convincingly have identified the peptide motif around amino acid T176 as target of proteasome-mediated degradation of GATA2 in benign basal type prostate cells. This is consistent with GATA2-degradation as published and referred to in some other cell types. Our attempts to reproduce previously published regulatory mechanisms according to which FBXW7 was the responsible E3 ligase targeting this peptide motif did not, however, come out so consistently, although reproducible. We consequently in this revision have moved the FBXW7 knockdown data to the Supplementary part. The review process has certainly helped us to improve the clarity of the manuscript regarding this, and we hope that the reviewer can appreciate this explanation.

This study indicated that the transcription factor GATA2, although unstable, may have different effects in TA cell types. Although exogenous GATA2 could be rescued by MG132-treatment, it was difficult to investigate further the physiological role played by GATA2 on AR transcription. One serendipitous discovery in this work was that MG132-treatment induced detectable AR mRNA in 957E/hTERT and EP156T cells and increases in PrEC cells in contrast to in untreated control cells. This work shows that proteasome inhibition in TA cells induces critical gene expression changes relevant for luminal differentiation, and this could represent a model for the identification of essential cell co-factors and molecular mechanisms of AR expression in benign prostate cells as one essential feature of luminal differentiaton. We consequently, in this revised version, has upgraded Fig. 4 with microarray data showing effects on a luminal transcript profile.

Reviewer 3: Fig 1. what is the rationale of studying the degradation of exogenous GATA2? Fig 1C, showed exogenous GATA2 protein degradation was E3 ligase dependent, and Fig 1E&F showed the endogenous GATA2 protein level was not increased after treating MG132. This cannot rule out the possibility of due to the exogenous overexpressed GATA2 protein is degraded by E3 ligase and the endogenous GATA2 protein is degraded differently. Fig 1D, increased AR mRNA level in 957E/hTERT cells are very low, and it is not altered by overexpression of GATA2 and treatment of MG132. This is very irrelevant to the hypothesis, please explain the rationale. Fig 1G, it is very hard to argue that it is a protective mechanism to exogenously overexpression of GATA2 which is not increased by MG132 treatment in 293 cells, in vitro. What is the rationale of proposing such mechanism of GATA2?

Author response: We appreciate and understand the comment. We should like to point out, however, that the conclusion of the experiment is that GATA2 in benign prostate cells seems to be very labile and subject to proteasome-mediated degradation, that can be inhibited by the proteasome-inhibitor MG132. The results show that GATA2 is well expressed, is relatively stable and GATA2 levels are not much increased by MG132 in 2 different malignant prostate cancer cell types Fig. 1E&F, in contrast to in benign prostate cells Fig. 1B. Exogenous expression of GATA2 in the non-transformed cell line HEK293 cells Fig 1G also appeared stable and was not much increased by MG132, suggesting that benign prostate cells differed from all the different control types regarding GATA2 stability. The results were highly reproducible. It would be difficult to distinguish between exogenous and endogenous GATA2 in LNCaP and 22Rv1 cells. We consequently hope that it can be accepted that this explanation supports our conclusion of very different regulation of GATA2 levels in benign prostate cells as opposed to both malignant prostate cells and another benign cell type.

Perhaps Fig. 1D shows the most important discovery of AR mRNA transcription in TA cells caused by MG132-mediated proteasome inhibition. The flow of data follows as: AR mRNA was first tested in Fig. 1D after the introduction of wild-type GATA2 in TA cells where the exogenous GATA2 was rapidly degraded. Although AR mRNA levels were found independent of wild-type GATA2 in Figure 1D but with the introduction of more stable mutant GATA2 T176A, AR mRNA level were checked again in Figure 3A.

Reviewer 3: Fig 2, it is not a good model to study E3 ligase activity in an exogenously overexpression system. And, the MG132 effect on GATA2 wt is not consistent between Fig 2A, and C, E, G, and there is no effect of MG132 on GATA2 T176A in Fig 2A, and C, E, G.

Author response: To investigate whether the exogenous expression of GATA2 in prostate epithelial basal-like transit amplifying cells could induce AR transcription and luminal differentiation, GATA2 wt was introduced in TA cells i.e., 957E/hTERT cells (Fig. 1) and EP156T (Fig. S1) to establish stable cell line but GATA2 protein showed rapid degradation. To investigate further if higher levels of GATA2 was able to induce expression of AR mRNA, in Fig. 2 TA cells were tested for GATA2 wt protein level along with more stable mutant GATA2 T176A immediately after transduction with or without MG132 to study the effect of GATA2 cofactor on AR mRNA transcription in Fig. 3.

We would also like to point out that we have updated the figures adding the quantification for all the western blots presented in the main figures. While Fig. 2 showed GATA2 overexpression, both GATA2 wt and GATA2 T176A proteins showed signs of degradation before being rescued by the treatment of MG132. The information obtained from the ratios in each lane supports the conclusions drawn.

Reviewer 3: Fig 3, the AR mRNA expression in Fig 3A is too low to have scientific significance.

Author response: We appreciate the comment and acknowledge the low levels of AR mRNA expression. In this section, the focus lies on our important discovery of detectable level of AR mRNA expression in TA cells following MG132-treatment irrespective of GATA2 wt or GATA2 T176A transduction. Only PrEC cells showed AR mRNA expression without MG132-treatment or GATA2 expression, so tested if the difference is significant following MG132-treatment and introduction of GATA2 wt or GATA2 T176A effects even in low levels. Both our group (1) and other groups have worked extensively in order to examine the mechanism of tightly restricted AR transcription in basal prostate cells and the very critical mechanisms that relieve this restriction and allows AR transcription and expression in order to co-ordinate a luminal and secretory program. This is well documented in a previous publication from our group (and other groups, references therein (1). We hope that this explanation will clarify this issue.

Reference:

  1. Olsen JR, Azeem W, Hellem MR, Marvyin K, Hua Y, Qu Y, et al. Context dependent regulatory patterns of the androgen receptor and androgen receptor target genes. BMC cancer. 2016;16:377.

Reviewer 3: Fig 4A. the FBXW7 silencing effect should be assayed by Western blot. And, the silencing effect is low in GATA2 wt and T176A overexpressed cells, that could related to the different expression of GATA2 mRNA expression in Fig 4B.

Author response: We thank the reviewer for the comment. The effect of FBXW7 silencing have been assayed upon multiple repeats by Western blot. We observed a detectable increase in GATA2 protein levels, but since the bands were very faint even by trying multiple times the figure with these data were moved to Supplementary section and not shown as a main result. Admittedly, these results are not clear, but they are consistent. The explanation is likely to be that several different mechanisms are operating at the same time as we have pointed out in the Discussion part. For that reason, now we have also moved Figs 4A and 4B to new Supplementary figure 2, thus showing the whole section of FBXW7 knockdown as supplementary information and not as our main results. The microarray gene expression heatmap, previously shown as supplementary figure 2, is now the new Fig. 4, emphasizing the effect of GATA2 on the induction of luminal marker genes in TA cells. 

We are thankful to reviewers to help us to clarify the manuscript and the main message as we think has been achieved in the currently revised version.

Round 2

Reviewer 2 Report

The authors have made additional changes that are globally in line with the reviewer's requests. Though some part of the study are not really convincing (including the fact that effects are quantitatively modest) the manuscript could be published in Biomedicines.

Reviewer 3 Report

Authors have addressed my concerns.

This manuscript is a resubmission of an earlier submission. The following is a list of the peer review reports and author responses from that submission.

Round 1

Reviewer 1 Report

The paper by Azeem et al. investigated whether ectopic expression of the GATA2 cofactor in basal-like Prostate cells could induce AR transcription and the luminal differentiation.

As a general remark, the manuscript is difficult to read and English editing would really facilitate to understand the logic of the experimental work.

Furthermore, as it is detailed further, the data that are presented cannot sustain the conclusions that are provided by the authors. Several crucial issues are as follows:

1- As a first remark, it is known for a long time that GATA2 directly promotes expression of both full-length and splice-variant AR (Proc Natl Acad Sci U S A. 2014 Dec 23;111(51):18261-6.) and that FoxA1 is part of the regulatory complex involved in this process.

Also, a negative feed-back loop showing that GATA2 expression was repressed by androgen was evidenced in the same study, which may explains the results of Table 1

2- Figure 1: First the western blots that are shown are never quantified, which does not give a sense on the level of increase that is observed. In fig. 1C, GATA2 increase I obvious following MG132 treatment. However, it is difficult to compare that increase with the increase shown in Fig. 1F for 22RV1 cells. Indeed, and conversely to what it is mentioned, there is a significant increase in GATA2 expression following treatment with 10 µM MG132…

Fig. 1D the Y axis scale goes form 0.1 to 10 ….please explain the absence of 1 ?

Fig. 1B, no legend on the side ? I supposed it is GATA 2 on the top…

3- I have a very hard time understanding the purpose of Fig. 1F? The text is not understandable and should be re-written in order to get the logic to use that cell line.

4- Figure 2 is raising several issues :

First, Western blots are not quantified which makes it difficult to appreciate the extent of the differences that are observed?

It is true that there is an increase of GATA2 for both WT and T176A mutant when 957E/hTERT cells are treated with MG132 and that this is accompanied with a significant difference in mRNA levels. It is also true that this phenomenon is also observed for WT GATA2 in other cell models but not for GATA2 TT176A mutant. However, as statistics are shown above the bar graphs, it is difficult to believe that differences in mRNA levels between WT and mutant GATA2 are relevant.

Fig. 1G should be re-probed for the same loading control as other blots (i. e. GAPDH) as actin bands are barely visible and show big variations.

5- Figure 3B is missing part of the legends. Please reconcile

6- Results of Figure 4 are also puzzling as siRNA effects that are shown in figure S2 are showing no repression of FBXW7 at all at the protein level. Since FBXW7 is one E3-ligase involved in the ubiquitination of GATA2, the protein level should be monitored.

Therefore the results of that figure are inconclusive. The authors should use other siRNA and must show a reduction of FBXW7 expression at the protein level to be able to interpret the results on GATA2 stability.

Reviewer 2 Report

In the present study, the role of GATA2 in the regulation of the AR mRNA and luminal AR target genes in cell culture models with a basal phenotype. The study is of interest, and luminal differentiation is an important research area. However, several issues need to be addressed before publishing. The following are my comments:

  1. Which isoform or transcript variant is detected in the used primers? Is there anything known about the role of the different isoforms?
  2. The group demonstrates that MG132 induces AR mRNA independent of GATA2. Is the observed effect on mRNA reproducible in other AR negative prostate cell lines (BPH1, PC3, PrEC, RWPE1/2)
  3. The relative AR mRNA seems very low. Are the detected Ct values still in the linear measuring range of the primers? Does gel analysis validate the values and an actual PCR product can be seen?
  4. Is there any explanation why the protein regulation is not affected by the T176A mutation?
  5. The data presented in Figure 3B should be validated on mRNA (by qPCR) and protein level. Also, labeling is missing in the figure
  6. Western blots of siRNA knock-down efficiency should be shown. “Data not shown” should not be avoided.
  7. Western Blots should be shown in Figure 4.
  8. Does the FBXW7 influence AR levels or luminal differentiation?

Author Response

Please see the attachment point-by-point response to the reviewers' comments.

Round 2

Reviewer 2 Report

As mentioned before, data not shown should be avoided. However, as the authors note the results in the manuscript, they add information. Therefore all results should be shown or not be mentioned.